# Ruminal Fistulation and Cannulation: A Necessary Procedure for the Advancement of Biotechnological Research in Ruminants

**DOI:** 10.3390/ani11071870

**Published:** 2021-06-23

**Authors:** Cristina Castillo, Joaquin Hernández

**Affiliations:** Unit of Animal Pathology and Clinical Examination, Department of Animal Pathology, Faculty of Veterinary Sciences, University of Santiago de Compostela—Campus of Lugo, 27002 Lugo, Spain; joaquin.hernandez@usc.es

**Keywords:** animal experimentation, ruminants, fistulation, cannulation, in vitro fermenters

## Abstract

**Simple Summary:**

This article addresses the role of ruminal fistulation and cannulation as an essential procedure in the advancement of research related to several items: fermentation in the ruminant forestomach, effects of new food sources, rumen diseases as well as the minimization of methane emissions, implicated in the so-called greenhouse gases. The aim is also to diminish the alarmist news promoted by animalist sectors, which accuse this technique of being an act of cruelty. This paper describes the importance of this procedure as a necessary in vivo tool for biotechnological research. In addition, we highlight the necessary management measures to ensure animal welfare. This review ends with a description of current in vitro methods as an alternative to in vivo studies, assessing their applicability as a complementary tool to the knowledge of rumen dynamics.

**Abstract:**

Rumen content is a complex mixture of feed, water, fermentation products, and living organisms such as bacteria, fungi, and protozoa, which vary over time and with different feeds. As it is impossible to reproduce this complex system in the laboratory, surgical fistulation and cannulation of the rumen is a powerful tool for the study (in vivo and in situ) of the physiology and biochemistry of the ruminant digestive system. Rumen fistulation in cattle, sheep, and goats has been performed extensively to advance our understanding of digestive physiology and development, nutrient degradability, and rumen microbial populations. The literature reports several fistulation and cannulation procedures in ruminants, which is not the focus of this paper. However, this method questions the ethical principles that alter the opinions of certain animal groups or those opposed to animal experimentation. In this article, we analyze the objectives of fistulation and cannulation of ruminants and the care needed to ensure that the welfare of the animal is maintained at all times. Due to the ethical issues raised by this technique, several in vitro digestion methods for simulating ruminal fermentation have been developed. The most relevant ones are described in this article. Independently of the procedure, we want to point out that research carried out with animals is obliged by legislation to follow strict ethical protocols, following the well-being and health status of the animal at all times.

## 1. Introduction

The complexity of ruminant digestion requires a greater variety and depth of experimental methods than that of any other species [1]. Ruminants have evolved to consume large amounts of fibrous plant material and rely heavily on the microbial breakdown of this feed in the rumen. Thanks to their ability to both harvest and digest complex carbohydrates prevalent in diverse locations and present in agricultural and industrial wastes, ruminants serve the population by converting useless and underutilized resources into food and fiber products that have high nutritional and economic value [2,3].

Ruminants have a stomach with four compartments or chambers—reticulum, rumen, omasum, and abomasum. The reticulum, rumen, and omasum are lined with non-glandular mucous membranes, while the abomasum is similar in function to the human stomach. The largest compartment is the rumen, which along with the reticulum serve as the sites of anaerobic fermentation. Ruminant nutritionists often refer to these compartments as the reticulo-rumen because together they function in the rumen cycle (coordinated contractions) to support the acts of eructation and rumination [2].

Rumen content is a complex mixture of feed, water, fermentation products, and living organisms such as bacteria, fungi, and protozoa, which vary over time and with different feeds [4]. As it is impossible to reproduce this complex system in the laboratory, surgical fistulation and cannulation of the rumen is a powerful tool for the study (in vivo and in situ) of the physiology and biochemistry of the ruminant digestive system (Figure 1). Rumen fistulation in cattle, sheep, and goats has been performed extensively to advance our understanding of digestive physiology and development, nutrient degradability, and rumen microbial populations [3,5,6,7].

The study of this complex environment has acquired special relevance over the years. As we will see later, knowledge of the fermentative processes that take place in the rumen constitutes a biotechnological tool that will help in the short term to understand physiological aspects of ruminants related to current challenges such as the search for new nutritional sources that do not compete with cereals for human consumption, as well as the minimization of greenhouse gas emissions. For these purposes, it will be necessary to study new nutritional sources in situ, assess their digestibility, and determine their influence on animal production. The fact is that studies performed concerning productive or metabolic functions do not always deal with the expectations created for the product being tested because the digestibility of a new food source is not always as expected.

But this technique is not only applied to adult ruminants; calves are also employed in the assessment of various feeding strategies and their effects on the ruminal environment and ruminal development [6]. Therefore, it is necessary to gain knowledge of the factors that alter some physical conditions or the chemical balance of the rumen, which improve production and the performance of these animals. However, differences in the composition of the ruminal community are not the same in all animals. There are differences attributable to diet, environment, health, animal genotype, and the age of the animals [8,9,10], so an experiment with a novel feedstock will not necessarily be valid on another farm in the world.

Historically, this procedure has its origins in 1822 with Dr. William Beaumont, who observed the secretion, motility, and emptying of the stomach in a patient with a gastric fistula, which was caused by a firearm injury after healing [11]. Decades later, in 1928, Schalk and Amadon described the technique of cannulation “in one stage”, for bovine and ovine, as simple, safe, and an ethical procedure, with minimal effects on or complications for the animal. From this technique, new variants have emerged, depending on the species [4,12]. Nevertheless, it is not the focus of this article to provide a description of each of them.

In a normally functioning rumen, fiber, starches, and sugars yield volatile fatty acids (VFAs), carbon dioxide (CO_2_), and methane. Protein sources are partially degraded to NH_4_, volatile fatty acids, and gases. Lipids are partly cleaved to glycerol and fatty acids, with unsaturated fatty acids being hydrogenated. Most organic compounds in the diet can be fermented by anaerobic microbes within the rumen. The extent to which feed components are degraded is limited either by the accessibility of various feed components to ruminal microbes, by the enzymatic activity of ruminal microbes (that will vary with ruminal conditions), or the amount of time available for fermentation [3]. Methanogens metabolize the hydrogen produced during fermentation. This microbial population is found free in the rumen fluid, attached to particulate material and rumen protozoa, as well as to the rumen epithelium. The methanogens associated with these different fractions can be expected to have different growth rates since they are removed from the rumen at different rates. For this reason, the animal itself and the feed influence the rate of passage of digesta through the rumen system [9].

But there are even studies [13] that point to the fact that cannulated rumen can potentially be utilized as a live laboratory for the investigation of the performance of antimicrobial materials associated with human and animal-related infections. This aspect is of great relevance, given the worldwide concern regarding the excessive use of antibiotics in animal production.

We can point out that nowadays, ruminal fistulation and cannulation have several utilities, which we will describe hereafter.

## 2. Current Uses of Fistulation and Cannulation in Ruminants

Taking into account the challenges that face ruminant livestock farming, this procedure can have several applications

### 2.1. The Search for New Nutritional Sources: Near the Area Where Livestock Is Raised and Can Replace Cereals or Forages Destined for Human Consumption

Of the world’s grains, one-third is destined for the feeding of livestock, 40% of which is destined to feed cattle. Although crop production increased by 47% between 1985 and 2005, in the face of the world’s population growth, there is stiff competition between food meant for human beings and that destined for livestock. Given the continuing increase in population, biotechnology has contributed to the modification of cattle nutrition in the last decades [8]. These widespread changes involve the use of by-products derived from agriculture, or even from the sea (algae, marine plants, and shellfish meal). However, not all of these potential foods have the same digestibility and can therefore negatively affect livestock production. The fact is that reality imposes the search for alternatives, but not at the expense of the profitability of the farm.

The technique of fistulation and cannulation allows for the analysis of dietary nutrient bioavailability through the in situ “nylon bag technique” (Figure 2), in which a feed sample contained within a bag of nylon filter-cloth is incubated directly in the rumen of a suitable live animal. A direct measurement of rumen digestion is provided, which can then be related to the period of incubation; this is less feasible with other in vitro techniques [13,14].

### 2.2. Control of Greenhouse Gas Emissions

One of the main problems associated with the rumen is the production of methane (Figure 3), which is carried out by methanogenic archaea and has been associated with the global warming phenomenon [15].

Limiting the activity of rumen methanogens in domesticated ruminants may result in gains in animal productivity if the rates and patterns of feed fermentation are not adversely affected. Interest in inhibiting rumen methanogens has recently been renewed due to concerns about the amounts of methane generated by domesticated ruminants. Knowledge of the ruminal methanogen community is a relevant part of the development of strategies for mitigating rumen methane production [16,17]. Assuming that the limited data available constitute a good sample of global ruminant archaeal diversity, it can be supposed that only specific groups of methanogens need to be targeted by antimethanogen agents initially. However, there is doubt whether the elimination of these groups will allow other lesser members to fill the vacant niche. Therefore, in vivo studies of all factors in the ruminal environment that limit or allow the coexistence of microbial populations are of great importance [9].

Different studies [17,18] highlight that this fact, rather than being a problem, is more of a possibility to generate renewable energy through specialized in vitro cultures. Through a stable and continuous artificial rumen system, rumen biomass from cellulose can be transformed into biogas.

The amount of methane expelled by the animal is directly related to the quantity (energy value) and quality (digestibility) of feed consumed. If the ration has good digestibility, the energy produced is used to increase weight and production in the form of meat/milk, thus decreasing the energy needed to form CH_4_, which in normal conditions is between 3% and 8% of the energy consumed [19]. Nevertheless, new questions, such as how the ruminal microbial groups coexist and determine the abundance of the different species, need to be answered.

### 2.3. Effect of the Ruminant Production System on Ruminal Balance

Related to the two previous points, the current predominant production models are the grazing system, more abundant in developing countries, and the intensive system, which predominates in developed countries. Both systems involve different types of feeding, with different productive results.

Pasture-raised animals are fed diets with low digestibility and nutrient content, which leads to decreased productivity and increased methane emissions. In this scenario, numerous studies have been conducted during the last decades on how to decrease or inhibit methane production. For this purpose, several options have been evaluated, such as the inclusion of lipids in the diet, the use of nitrates, ionophores, tannins, and alkaline treatments. However, the final decision is closely related to herd productivity [19].

On the other hand, intensively raised animals are fed highly digestible and nutrient-rich diets, which favors productivity. However, the economic demands of this farming model have caused them to consume an increasing amount of cereals in order to achieve rapid growth. This nutritional model has provoked the appearance of metabolic diseases such as ruminal acidosis, bloat, or ketosis, or even emergence of public health concerns related to the use of growth promoters linked to these diets, such as the antibiotic monensin. Knowledge of the etiopathogenesis of these diseases as well as the search for alternatives to monensin have recently implied the need to study in vivo what happens in the rumen [12,20] and assess its biochemical and microbiological changes.

### 2.4. The Search for Solutions to Ruminal Diseases

Both rearing systems have their peculiarities. For example, in grazing animals, the use of herbicides such as glyphosate can alter the fungal community in the rumen of dairy cows, leading to imbalances (sometimes called “dysbiosis” or “dysbacteriosis”) with subsequent clinical signs [21]. Ruminants that are rearing under intensive conditions have the same problems, which in this case is related to the imbalance in normal ruminal population due to the increase in grain consumption, which modifies rumen pH.

In both situations, rumen microbial populations are low or are inappropriate for the diet being consumed. It then becomes necessary to replace the rumen fluid of these animals with impaired rumen digestion with that which contains microbes and nutrients from healthy animals. This process is called “transfaunation”. For this process, a cannulated donor animal can be a long-term, readily available source of rumen content that can be used to transfaunate herd mates that have suffered various primary or secondary digestive upsets [2,22,23]. There are researchers who point out that as long as cannulated and fistulated animals are kept in correct hygienic conditions, they can remain in this state for 3–4 years without the appearance of clinical complications [12,24].

Although so far we have only focused on the nutritional, productive, and environmental aspects that this technique helps to elucidate, it also allows us to comprehend key aspects associated with digestive infections such as the one caused by *Clostridium perfringens*, which is frequently present in the bovine rumen and causes hemorrhagic enteritis in calves, enterotoxemia, jejunal hemorrhage syndrome, abomasal ulcers, and tympanic or gaseous gangrene [25].

### 2.5. The Study of Rumen Microbiota and Microbioma

As described in previous paragraphs, sustainable animal farming aims towards the consumption of local resources. Therefore, new techniques are being developed and implemented to improve the rumen fermentation process. Rapid advances in molecular biology and phylogenetics as well as the rise of “omics” approaches have provided insight into the ecology and function of microbial ecosystems in the rumen [26].

Bacteria predominate in the rumen microbiota and are responsible for the conversion of non-digestible plant biomass into energy and the formation of microbial protein. Both processes determine the productive efficiency of ruminants. Elucidation of the interactions between microbial populations has the potential to improve production. Recently, next-generation sequencing technology has enabled the sequencing of microbial genomes in a relatively short period. There are different high-throughput sequencing platforms on the market that have been applied, improving our knowledge of rumen microbes, their genes, and enzymes [8,27].

Therefore, regular collection and analysis of rumen samples from ruminants are necessary for investigating the composition of the rumen microbiome, which contributes to the effective digestion of plant materials and rumen fermentation [28]. Currently, we know that in cattle, the most abundant phyla are *Bacteroidetes,* followed by *Firmicutes*. Less abundant are *Fibrobacteres*, *Proteobacteria* and *Tenericutes*, *Actinobacteria*, and *Spirochaetes* [29]. Despite extensive experience in studies on rumen microbiota in cattle, attempts to manipulate rumen fermentation continue to yield short-term results in sheep and goat species [30].

As we can see, there are several reasons why this in vivo and in situ procedure continues to maintain its relevance in clinical or experimental studies in ruminants in which the digestibility of new nutritional sources, the influence of the ration on methanogenesis, or the solution of digestive disorders needs to be known in detail. The problem is that the appearance of this type of experimental animals in the media, for inexplicable or intentional reasons, creates unnecessary alarm in a society that is unaware of what these studies are about and encourages the attitudes of animalist groups or those opposed to animal experimentation itself.

A lot of people are unaware that research carried out with animals is obliged to follow strict protocols that assess the real need for the procedure, the possibility of using replacement methods and the minimum number of animals, and especially if the procedure considers the well-being and health status of the animal at all times. All this is in accordance with the European Union and national regulations for animal experimentation that adopted the Directive 2010/63/EU on 22 September 2010, which entered into full effect on 1 January 2013 [31].

Considering the strict measures required by law for implementing this procedure, during which animal suffering has to be extremely minimized, the usefulness of fistulation and cannulation is thus in the interest of human beings in several aspects:The search for sustainable livestock farming by taking advantage of local resources, and especially by using vegetable by-products that do not affect digestibility and rumen balance. For this, preliminary research through fistulation and cannulation is an essential step. These by-products, rich in antioxidants, contribute to what we know today as “food fortification”, adding value to the final product (milk/meat) through supplements of natural origin. In addition, through the constant control of antibiotics abuse in livestock farming, it has been demonstrated that antioxidants of natural origin have an antibacterial effect.From an environmental point of view, the control of greenhouse gas emissions has motivated the development of research in search of nutritional sources that minimize them, as well as the genetic study of the microbiome. Objective results cannot be obtained without in situ access to the rumen chamber.Finally, the study of rumen diseases, especially infectious diseases, contributes not only to the control of the farm’s economic losses but also to the prevention of such diseases, or even to the inhibition of new pathogens that end up becoming what is currently called “emerging diseases of animal origin”, which currently has such a great repercussion in the media as a result of the COVID-19 pandemic.

## 3. The Process of Ruminal Fistulation and Cannulation

At least three different ruminal fistulation techniques are used in ruminants: a one-stage method, a two-stage method, and the Schalk and Amadon technique [6,32]. The first technique is suitable for the implantation of small cannulae and is generally applied in small ruminants, whereas the second is preferred in larger ruminants. However, this criterion is not definitive since the use of one technique or the other will depend on the objectives pursued by the study [5]. Every year it is possible to read articles that make modifications to the previously mentioned techniques and that allude to the surgical approach, the type of cannula to be used, or even the strategies to be followed after surgery to maintain the anaerobic environment of the rumen. Regardless of the technique or location, the gap between the skin and rumen cannula can lead to fermentation gas leakage and atmospheric air ingress, which can negatively affect the anaerobic environment of the rumen; therefore, the choice of the appropriate material according to the species and/or the size of the animal is an aspect to be taken into account [27]. From reading the available references, we would like to highlight two facts: (1) that a good surgical procedure allows for the long-term use of cannulated animals [33], and (2) that this procedure is impractical for sampling a large number of animals [28,34].

For microbiological studies, rumen fluid can be obtained from the dorsal or ventral sac of the rumen [28]. Other authors [20] consider that rumen pH varies significantly among sites within the rumen. This fact is relevant when it concerns the study of the rumen fermentability of new nutritional sources and their effect on rumen physiology and biochemistry. According to this criterion, and to prevent the occurrence of ruminal acidosis, cannulation and fistulation should take place in four sites: cranial ventral rumen, caudal ventral rumen, central rumen, and cranial dorsal rumen, with the cranial-ventral rumen being the better place as it is where most mixing of rumen contents occurs and rumen pH is more reliable.

In general terms, rumen cannulation is performed on a healthy animal with minimal expense. The surgical method used is not more complicated than the one that is always carried out by veterinarians for other procedures in cattle.

The first and probably the most critical aspect of experimental success is the selection of the animals. Their temperament is essential. Ideally, before surgery, the animals should be trained for the restraint and handling involved in the sampling. They will also have to get used to the space where they will go for cannulation, evaluating the dangers that may exist such as the walls of the facility—which must be smooth, the openings in the fences or doors—which may damage the animal or trap and pull out the cannula, the possibility of the cannula being ejected when the animal is moved or lifted, or even the free access of other animals that may pull out the cannula. Additionally and obviously, preventing its exhibition to people who are unaware of the purpose of this procedure and the strict adherence to protocols. Often, the lack of design forces surgery on animals that become excited and consequently damage themselves or the cannula after surgery [1]. The surgical site and the surface of the outer edge of the cannula next to the skin should be cleaned daily for 5–7 days with a diluted antiseptic solution. The wound should be protected from flies, and a broad-spectrum antibiotic must be administered after surgery for seven days. The use of postoperative analgesics if the animal appears to be uncomfortable is recommended [22].

Animal welfare includes not only adequate conditions in the experimental pen, but also the enrichment of the environment to improve their emotional state. A clear and concise description of the procedures used, and the care received by the animal, in our opinion, constitute a valuable tool that adds value to the experiment from an ethical point of view since it demonstrates that the researchers subscribe to an ethical position on animal experimentation [31,35]. On numerous occasions, and contrary to popular belief, the cannulated animals are subjected to reconstructive surgery at the end of the study. This allows for the removal of the cannula and the closure of the layers of tissue by planes; animals are then incorporated into productive and reproductive life within their herd of origin, thus avoiding their sacrifice [5].

The search continues for new procedures to reduce stress and post-surgical complications allowing the collection of reliable data. After all, if the animal is stressed by pain or mishandling, the samples could be altered, and the experiment will be unsuccessful.

Unfortunately, it must be recognized that not all research centers or universities maintain this criterion at present, so it is not out of place to continue emphasizing good practice and denouncing bad methods, as noted by Nature Journal editorial in 2013 (Volume 504, Failure of care: Nature News & Comment).

### How Do Fistulization and Cannulation Affect the General Health and Production of the Animals?

In young calves, the surgical technique has no major effect on animal health and performance, on the intake of milk or solid feed mixture, on body weight gain, and the voluntary intake of feed in comparison with other normal calves at similar ages [6].

A study conducted on dairy cows, assessing the effects of feeding wheat or corn on methane emissions [36] using rumen fistulation concluded that this procedure did not affect dry matter intake, milk production, or milk composition.

In sheep, Sharman et al. [7] compared the two main cannulation methods currently used in small ruminants (one-stage versus two-stage) and concluded that although there was no difference in vital signs, the two-stage method was more stressful. Previous studies [32] confirmed that no post-surgical complications (bleeding, local inflammation, infection, wound dehiscence or suture abscess at the surgical site) occurred with the one-stage technique. Mean values of physical examinations (rectal temperature, respiratory rate, heart rate, ruminal movements) were within physiological limits. The animals maintained their body weight in the first week after surgery and then gradually gained weight. Similar results were obtained in goats [37].

Although a good procedure does not imply any risk to the animal’s health [38], some exceptional cases are usually solved with a rumenotomy. After the procedure, monitoring of vital signs is the standard and simple way to assess the animal’s general health status. An abnormally high body temperature (pyrexia) may indicate the presence of infection.

Kebamo [38] describes the case of a cow subjected to fistulization. The affected animal showed ruminal discharges from the injured site. On examination, the rumen wall was found injured and adhered to the flank wall at the wounded area. Hence, treating traumatic ruminal fistula by rumenotomy with good postoperative management could be considered a successful surgical procedure.

## 4. Alternatives to Rumen Fistulization and Cannulation

### 4.1. In Vivo

As an alternative to microbial community analysis by ruminal fistula, other less aggressive methods have been used over the years. Rumen sampling can be carried out by oral intubation, but this is an unpleasant procedure for the animal and also results in a sample that is often heavily contaminated with saliva [39]. For this reason, recommendations for obtaining rumen fluid with this technique have included discarding the first 200 mL of fluid obtained [20,28].

Rumenocentesis provides valid samples but involves puncturing the abdominal wall with a needle and removing digesta by syringe, also undesirable in terms of animal welfare [20] and restricts the amount of sample that can be collected. A study performed by Duffield et al. [20] in dairy cows comparing oro-ruminal sampling, rumenocentesis, and cannulation demonstrates that the last was the best at showing the biochemical changes taking place in the rumen.

Other methods could be the evaluation of feces, or regurgitated digesta (bolus). Briefly, ruminants regularly regurgitate rumen contents to chew partially digested plant material [40]. The chewed bolus is then swallowed for further microbial degradation. Therefore, it may be assumed that the microbiota of the mouth could represent a reflection of rumen microorganisms. Thus, the collection of small buccal fluid samples could be used as an indicator to assess the microbial ecology of the rumen, avoiding invasive procedures. Concerning feces, although its microbiota is significantly different from the rumen microbiota [10,15], there are indicators such as fecal *archaeol*, a membrane lipid of rumen archaea that has been found to be a useful marker of rumen methanogenesis [41].

An interesting study published by Tapio et al. [34] tried to compare the communities of these alternative samples from bovine in order to evaluate their usefulness as substitutes for the direct sampling of rumen digesta, as had already been tested in sheep [42]. The conclusions obtained in the study reflect that the collection of salivary samples is not equal to of the collection of rumen samples. The archaea:bacteria ratio in oral sampling was different from the corresponding rumen samples. Nonetheless, unlike rumen fistulization, this method appears to be useful for screening purposes in large animal populations or herds. Finally, the microbial composition of the feces did not represent the rumen digesta and has no value as a biomarker of rumen function.

### 4.2. In Vitro: The Use of Fermenters

Due to the labor and ethical problems posed by the use of fistulated and cannulated animals to examine the digestive tract, several in vitro digestion methods have been developed to simulate ruminal fermentation. However, cell culture-based studies that have been conducted in the last 50 years do not reveal the microbial diversity of the rumen as some species are easier to culture than others, and the sample size is often too small to provide a complete insight into the composition of the rumen microbiota [9].

The Rusitec, an acronym for “Rumen Simulation Technique”, is a well-established in vitro method to simulate and investigate rumen microbial processes, avoiding animal variability in a standardized environment [43]. This method is widely used to study the effects of different diets or feed additives on microbial fermentation pathways, protein synthesis, and microbiome growth [44]. Despite being a highly standardized method (e.g., in terms of temperature, pH, and buffer flow) the system is known to differ from in vivo conditions in terms of absorption processes, differences in the ratio of liquid to solid materials, lower concentrations of short-chain fatty acids (SCFA), and protozoan shifts as compared to the donor animal [45].

Since the internationalization of Rusitec as the standard technique for in vitro evaluation of rumen fermentation processes, a lot of modifications have been developed in recent years that try to estimate feed digestibility and energy content more reliably (Figure 4).

However, despite all the attempts to improve this in vitro technique, it has become evident that there is a need to develop a system capable of automating the traditional in vitro digestibility analysis and solving some analytical errors, such as those related to sample handling and the manual filtration steps. A recent technique such as the Ankom DaisyII incubator [46] has gained acceptance as an alternative to traditional in vitro procedures.

This method was developed to predict the digestibility of feedstuffs for ruminants and has been modified and adapted to improve its accuracy and prediction capacity. Modifications used by various researchers include the use of different inocula, buffer solutions, and sample weights. However, as with Rusitec, for it to function, it needs material donated by another ruminant, which has to be fistulized and cannulated (Figure 5).

Finally, for industrial purposes, a “dynamic membrane bioreactor” has been developed to generate biomethane from lignocellulosic biomass (corn straw, wheat straw, rice straw, etc.) through in vitro fermentation of cow and sheep inoculums [17]. The novelty is that it can be used for biofuel production while maintaining a high density of rumen microorganisms, preserving them for months without losing much of their activity. This fact solves the problems with microorganism viability reported in previous techniques.

This system was evaluated by comparing VFA production, cellulose, hemicellulose, and lignin degradation efficiency, changes in the main lignocellulose degrading enzymes, and the characteristics of microbial communities (bacteria, fungi, and archaea) between bovine and ovine inocula. Furthermore, anaerobic digestion performance was compared based on the feed received (corn straw or food waste) after pretreatment of the rumen fluid and permeabilization of the artificial rumen systems (Figure 6).

According to the results obtained, the study conducted by Xing et al. [17] conclude that there was greater diversity and richness of bacteria and fungi in the bovine inocula, indicating that it is the most suitable for in vitro studies. In addition, the authors consider that the use of an artificial cow rumen system with dynamic membrane technology is a promising way to build a stable and continuous artificial rumen system.

## 5. Conclusions

Rumen fistulation and cannulation is an essential tool for the progress of ruminant research regarding the study of new food sources, particularly in the evaluation of their productivity, health status, or the greater or lesser potential for greenhouse gas production. It must be carried out with few animals and subjected to strict clinical and management controls that guarantee their welfare at all times. The use of in vitro fermenters does not replace the data provided by the live animal but can give additional information about the changes that take place in the rumen environment under standard conditions, independently of the animal. The latest developments are targeted towards the development of a stable and continuous artificial rumen system that allows for a better understanding of rumen dynamics with as few animals as possible.

Despite this, there are still sectors of the population that are reluctant to fistulation and cannulation in ruminants, partly due to the lack of knowledge of the technique, which must be performed with a minimum number of animals and taking into account their welfare and their health status, according to the requirements established by legislation.

## Figures and Tables

**Figure 1 animals-11-01870-f001:**
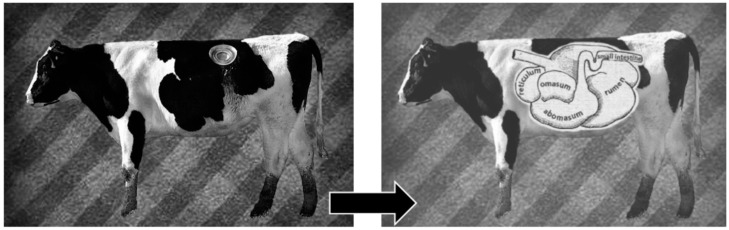
Cannulated cow and the anatomical site where the cannula is located (from University of California Research– How a permanent hole in a cow’s stomach is beneficial. From: https://ucresearch.tumblr.com/post/123651056610/the-expression-holy-cow-may-be-quite-appropriate, accessed on 15 April 2021).

**Figure 2 animals-11-01870-f002:**
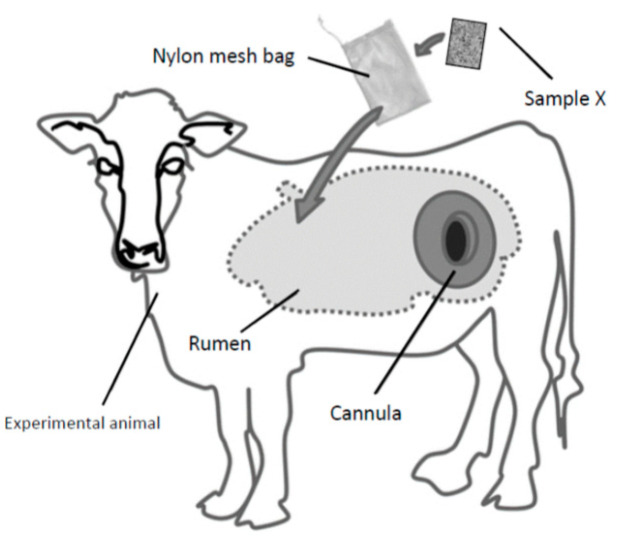
Schematic image of the in situ nylon bag technique (adapted from Berean et al., 2015 [13]).

**Figure 3 animals-11-01870-f003:**
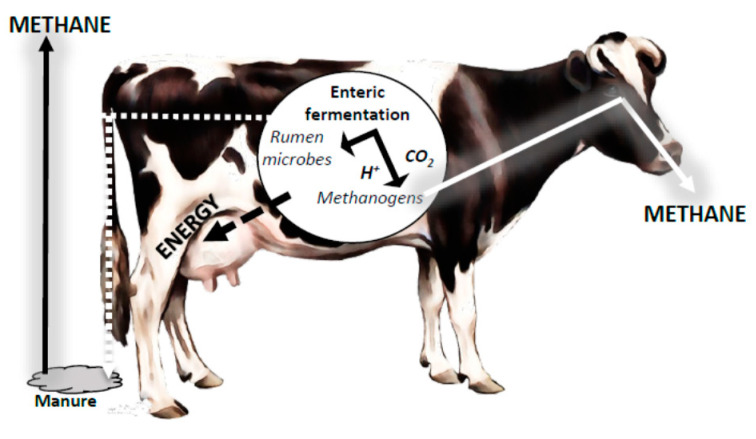
Schematic representation of the fermentative processes occurring in the rumen (adapted from Kumari et al. [16]).

**Figure 4 animals-11-01870-f004:**
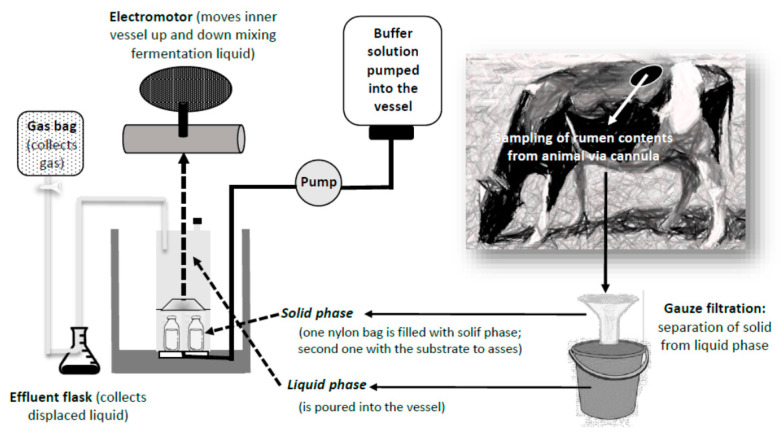
One fermentation vessel of the Rusitec system (adapted from Riede et al., 2006 [21]).

**Figure 5 animals-11-01870-f005:**
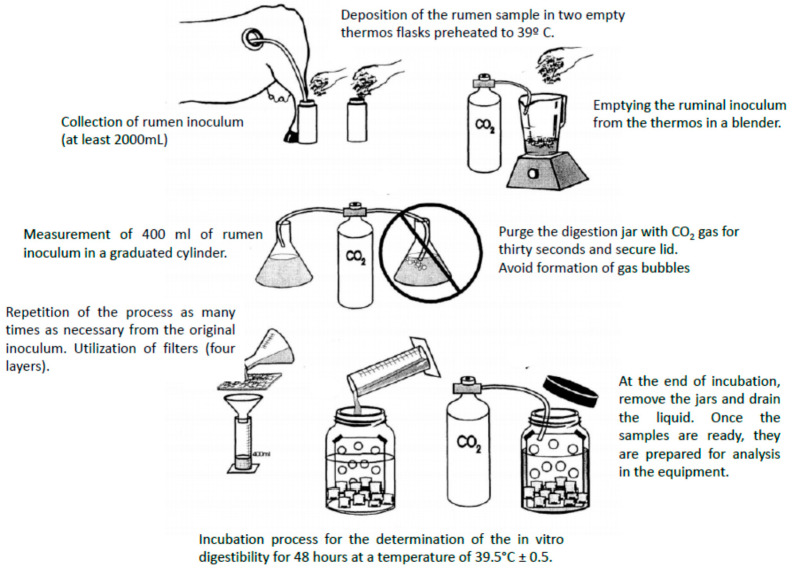
Schematic procedure for collection and preparation of rumen samples to be analyzed by the fermenter DaisyII Incubator (adapted from https://www.ankom.com/sites/default/files/document-files/Method_4_InVitro_Procedure_Illustrations_D200_D200I.pdf, accessed on 26 April 2021).

**Figure 6 animals-11-01870-f006:**
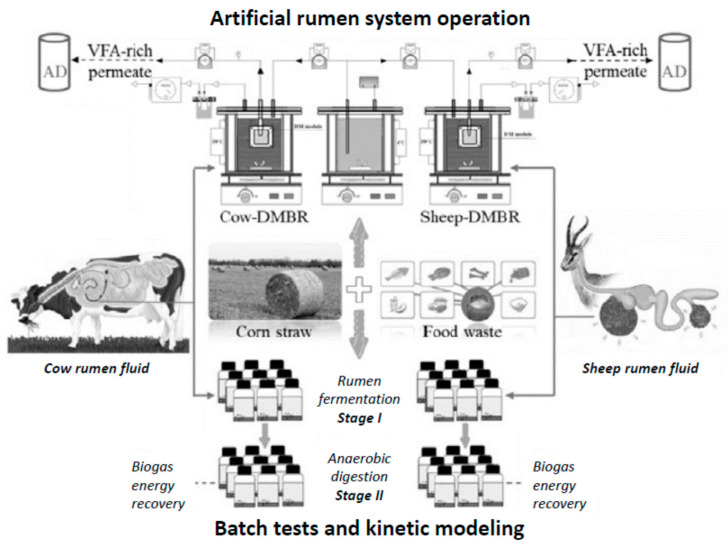
Artificial cow rumen system as a promising way to degrade lignocellulosis biomas recycling methane emissions and allowing a better characterization of ruminal population (from: Xing et al. [17]. Abbreviations: VFA: Volatile Fatty Acids; AD: anaerobic digestion; DMBR: dynamic membrane biorreactor).

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
