# Peer review of "Ruminal Fistulation and Cannulation: A Necessary Procedure for the Advancement of Biotechnological Research in Ruminants"

_animals, 2021, doi:10.3390/ani11071870_

Round 1

Reviewer 1 Report

Thank you for responding to the comments I presented in the first review and for making additions and corrections to the article. I also suggest ordering the numbering of subsections in the article. In chapter 2, individual subsections should be labeled 2.1, 2.2 etc. instead of 1, 2 ... which is confusing. There is only one section 3.1 in Chapter 3. Meanwhile, the essence of introducing subsections is that there must be at least two subsections to justify their application. Therefore, in the structure of chapter 3, one more subsection must be added, or the numbering 3.1 must be removed.

Author Response

REVIEWER 1.
Thank you for responding to the comments I presented in the first review and for making additions and corrections to the article. 
We are grateful for their appreciation. We would also like to point out that thanks to the changes prompted by the referee's comments, the article has been significantly improved.
I also suggest ordering the numbering of subsections in the article. In chapter 2, individual subsections should be labeled 2.1, 2.2 etc. instead of 1, 2 ... which is confusing. OK!
There is only one section 3.1 in Chapter 3. 
Meanwhile, the essence of introducing subsections is that there must be at least two subsections to justify their application. Therefore, in the structure of chapter 3, one more subsection must be added, or the numbering 3.1 must be removed.
The referee is right in this respect and we have changed the numbering of the subsections in the article, both in chapter 2 and 3

Reviewer 2 Report

I think that this paper has been revised based on the points raised.

There is only one factual misunderstanding that needs to be corrected.

P4L132  and in the destruction of the ozone layer

Methane is not a causative agent of ozone depletion, so delete this part.

That's all.

Author Response

REVIEWER 2
I think that this paper has been revised based on the points raised. Thanks in advance for the comments
. We would like to point out that thanks to the changes prompted by the referee's comments, the article has been significantly improved.
There is only one factual misunderstanding that needs to be corrected.
P4L132  and in the destruction of the ozone layer. Methane is not a causative agent of ozone depletion, so delete this part. 
We accept the referee´s comment. We have detleted that part of sentence. (see L-132)

This manuscript is a resubmission of an earlier submission. The following is a list of the peer review reports and author responses from that submission.

Round 1

Reviewer 1 Report

In the chapter Introduction, it would also be worth paying attention to the impact of animal production, especially dairy and beef cattle production, on the natural environment. It would be worth developing, at least in a short paragraph, the issues of harmful gases (methane) released into the atmosphere as a result of cattle production, and indirectly the specificity of the structure of their stomach / digestive system. Thus, the topic discussed in the article could be even more clearly justified by linking the considerations on rumen fistulization with the currently important environmental protection.

In the case of the presented Figures 1 and 2, it might be worth mentioning their source or on the basis of which sources they were developed.

I think that in the presentation of the current knowledge on rumen fistulization and its discussion, it would be worth taking into account the thread of connections between fistulization and the aspect of dairy cattle reproduction on the farm. Many years ago, while working on a dairy experimental farm, I had contact with dairy cattle in an experiment with fistulization. Although they were adult cattle, they were not inseminated. Perhaps it would be useful to elaborate on this in the discussion, taking into account aspects of the physiology of dairy cattle.

While staying in an experimental dairy farm where experiments with cattle with fistulization were carried out, such animals were presented at animal exhibitions. I think it would be worth writing about in an article whether showing fistulized dairy cattle is ethical both from the point of view of the animals and the society that such animals can view.

In the summary of the article, or possibly in the Conclusions, the authors could have their own opinion on the direction of further changes / approach to rumen fistulization in experiments with ruminants. Will it be more important to approach the methodical / conceptual, ethical, or maybe related to the search for technical solutions that will allow even more perfect experiments simulating phenomena occurring in the rumen of ruminants? A few sentences of the authors' own reflections would significantly increase the value of the concluding part of the article.

Reviewer 2 Report

The intention for submitting a manuscript to insist on the utility and necessity of fistulation for research progress is well understood. However, for the following reasons, I would like to recommend that you submit your work in another suitable journal, etc.

・The target to appeal to is not the main reader of this journal. If you want to educate the public, you should choose another journal.

・The content of this manuscript is well known to animal scientists, and novelty and originality are not recognized as grounds for asserting its usefulness and necessity.

・In order to enlighten the general public, I think it is more important to emphasize the necessity of nutritional research on ruminants than to insist on the necessity of the experimental procedure.

Reviewer 3 Report

Dear authors,

The review subject is very relevant to our field. However, the authors couldn't deliver it. The introduction does not bring the main topic, and the subsequent topics are not very well connected. In general, the authors should exploit more the published data in the world. 
For instance, reports regarding other gas production in vitro technics since Rusitec is not the only technic that uses rumen fluid from fistulated animals. We have significant articles published using GIV technic predicting in vivo methane and total gas emissions, which are only possible using fistulated donors. In situ evaluations such as iNDF and fiber degradation were not even mentioned.  Very relevant data is available in the literature but was ignored by the authors. 
Further, the authors concluded the article by saying, "in situations of digestive infection, fistulization and subsequent analysis in vitro fermenters will make it possible to identify the pathological agent and seek the best treatment". I don't think it is reasonable to cannulate an animal to treat it—unclear approach. 

The current form of the manuscript is shallow. Unfortunately, my recommendation is to reconsider after major revision. I would be thrilled to review a resubmitted version after improvements. 

Important: Title - fistulization is not right. 

Intrigued by the sentence in L95

L95 -  Why are you excluding Animal Scientists? I don´t understand it since animal science leads massively the research regarding livestock production, efficiency, welfare, and food safety.